# MINIMALLY REDUNDANT LAPLACIAN EIGENMAPS

**David Pfau & Christopher P. Burgess**
DeepMind
London, UK
{pfau, cpburgess}@google.com

## ABSTRACT

Spectral algorithms for learning low-dimensional data manifolds have largely
been supplanted by deep learning methods in recent years. One reason is that
classic spectral manifold learning methods often learn collapsed embeddings that
do not fill the embedding space. We show that this is a natural consequence of
data where different latent dimensions have dramatically different scaling in ob-
servation space. We present a simple extension of Laplacian Eigenmaps to fix
this problem based on choosing embedding vectors which are both orthogonal
and *minimally redundant* to other dimensions of the embedding. In experiments
on NORB and similarity-transformed faces we show that Minimally Redundant
Laplacian Eigenmap (MR-LEM) significantly improves the quality of embedding
vectors over Laplacian Eigenmaps, accurately recovers the latent topology of the
data, and discovers many disentangled factors of variation of comparable quality
to state-of-the-art deep learning methods.

## 1 INTRODUCTION

Classic nonparametric manifold learning methods like Laplacian Eigenmaps (LEM) (Belkin &
Niyogi, 2003), Locally Linear Embedding (LLE) (Roweis & Saul, 2000) and Isomap (Tenenbaum
et al., 2000) have many attractive properties. Learning is usually accomplished by a spectral decom-
position, which means a globally optimal solution can be found, and nonparametric methods usually
exhibit greater data efficiency than parametric approaches, especially neural networks. These meth-
ods are also able to learn a latent space embedding without having to reconstruct the output - they
are *inference* models rather than *generative* models. However in practice, these methods often suffer
from issues of collapsing embeddings, where the data are mapped to a thin manifold of even lower
dimension than desired (Hadsell et al., 2006). This is one limitation that has led to spectral manifold
learning methods being supplanted by other methods, such as Variational Autoencoders (Rezende
et al., 2014; Kingma & Welling, 2013) and Generative Adversarial Networks (Goodfellow et al.,
2014).

We argue that this collapse is due to the method by which embedding vectors are chosen. When
data are scaled differently along different dimensions, the smallest (or largest) eigenvectors of the
Gram matrix may simply be higher-frequency modes along the same direction rather than a low-
frequency mode along a different direction in latent space. By properly choosing the embedding
vectors to be *unpredictable* by the other vectors as well as orthogonal to them, we fix this issue
of embedding collapse. Interestingly, the improved embeddings often are aligned with naturally
disentangled factors of variation in the underlying data, suggesting that the results of recent deep
learning methods that purport to discover the same disentangled factors (Higgins et al., 2017; Chen
et al., 2016) may be reproducible by simpler, classic machine learning methods.

### 1.1 LAPLACIAN EIGENMAPS

Laplacian Eigenmaps (LEM) (Belkin & Niyogi, 2003) is a spectral method for mapping high-
dimensional data to a low-dimensional space. The algorithm is quite simple. First, construct a
symmetric adjacency matrix $A$ where $A_{ij} = 1$ if data $\mathbf{x}_i$ and $\mathbf{x}_j$ are neighbors. More fine-grained
information such as the RBF kernel between $\mathbf{x}_i$ and $\mathbf{x}_j$ can also be used, as long as $A_{ij} \geq 0$. From
this, construct the diagonal degree matrix $D$, $D_{ii} = \sum_j A_{ij}$ which gives the number of edges into

the node $i$. The graph Laplacian is then given by $L = D - A$, and the normalized graph Laplacian by $\tilde{L} = D^{-1/2}LD^{-1/2} = I - D^{-1/2}AD^{-1/2}$. When the underlying graph is a $d$-dimensional grid, the graph Laplacian can be seen as a discrete approximation to the Laplace-Beltrami operator (the "diffusion operator"), and thus for a distribution $\mathbf{p} = (p_1, \ldots, p_n)$ over nodes of the graph, $\dot{\mathbf{p}} = L\mathbf{p}$ is a discrete approximation to the diffusion equation. The Laplacian has a trivial uniform eigenvector with eigenvalue zero, corresponding to the stationary distribution of this diffusion, and the remaining eigenvalues are positive. Laplacian Eigenmaps uses the next $k$ eigenvectors with smallest eigenvalues as the $k$ dimensional embedding vectors for all data in the dataset. Other methods like LLE (Roweis & Saul, 2000) and Isomap (Tenenbaum et al., 2000) are closely related, primarily differing in how the Gram matrix is constructed (Bengio et al., 2004).

## 1.2 THE ISSUE WITH SCALING

Consider an $m$-by-$n$ grid. The eigenvectors of the graph Laplacian will be products of Fourier modes along different directions, where the eigenvalues depend on the values of $m$ and $n$. This is illustrated in Fig. 1. If $m \gg n$, then several higher-frequency modes along the first axis of the grid will have lower eigenvalues than the lowest mode along the second axis. Thus, if we simply pick the nontrivial eigenvectors with the lowest eigenvalues as the embedding, we will get an embedding that collapses along a single dimension. This is exactly what is observed in practice as noted by Hadsell et al. (2006), who showed that on subsets of NORB and MNIST, LLE often collapses in such a manner. This is likely due to the fact that different changes to the latents (e.g. pose and lighting) cause different scales of changes in observation space, so the nearest neighbors graph is more elongated along certain dimensions than others. Modern deep unsupervised methods like VAEs and GANs reduce the severity of collapsed embeddings by forcing the latent codes to be as close to a unit-variance Gaussian distribution as possible - a stronger requirement than just requiring different dimensions in latent space to be linearly uncorrelated.

## 2 MINIMALLY REDUNDANT LAPLACIAN EIGENMAPS

Here we propose an improved criterion for selecting eigenvectors for a low-dimensional embedding. The main idea is that, given $d$ dimensions of an embedding, the $d+1$st dimension should not only be linearly uncorrelated, but not predictable from the first $d$ dimensions using *any nonlinear function*. It is easy to construct vectors which are linearly uncorrelated but still predictable - for instance, given samples $x \sim \text{Unif}(0, 2\pi)$, $\cos(2x)$ has zero correlation with $\cos(x)$ and $\sin(x)$ but is still a deterministic function of them. Including such a vector in a low dimensional embedding would be superfluous.

Many different criteria could be used to quantify the degree of redundancy of a vector of data to other vectors. Given vectors $\mathbf{v}^1, \ldots, \mathbf{v}^{d+1} \in R^n$ of samples from random variables $V_1, \ldots, V_{d+1}$, we could use nonparametric nearest neighbors estimators (Beirlant et al., 1997; Kraskov et al., 2004) to estimate the conditional entropy $H[V_{d+1}|V_1, \ldots, V_d]$. These estimators depend on the log of the distance to the nearest neighbors, which tends to be highly sensitive to small distances, while we are more interested in large deviations from perfect predictability. We could also fit a Gaussian process which maps $\mathbf{v}^1, \ldots, \mathbf{v}^d$ to $\mathbf{v}^{d+1}$ and compute the likelihood of the data under such a prior, but this involves log determinants of large matrices and is likely to be slow.

Instead, we use a simple, robust heuristic inspired by nearest-neighbors entropy estimators, but more sensitive to deviations from exact predictability. We define "unpredictability" as follows: let $\bar{\mathbf{v}}_i^d = (v_i^1, \ldots, v_i^d)^T$ contain the values of the first $d$ embedding vectors at index $i$, then let $i_1, \ldots, i_k$ index of the $k$ nearest neighbors of $\bar{\mathbf{v}}_i^d$, and let $v_i^{d+1}$ be the value of the $i$th index of $\mathbf{v}^{d+1}$. Then a vector is considered "sufficiently unpredictable" if $\frac{1}{\sqrt{n}}\sum_i \sum_{j=1}^k ||v_i^{d+1} - v_{i_j}^{d+1}|| > \epsilon$. While this is not a correct estimator for any measure of predictability that we are aware of, we believe its simplicity, robustness and speed makes it an attractive choice.

To construct an embedding from a graph Laplacian, we first precompute the $D$ eigenvectors with smallest eigenvalue. We initialize the embedding with the eigenvector with second-lowest eigenvalue, then iterate through the remaining eigenvectors, selecting those that are sufficiently unpredictable given the rest of the embedding so far for inclusion. In our experiments a threshold $\epsilon$ between 0.3 and 0.5 usually worked well.

## 3    EXPERIMENTS

We evaluated Minimally Redundant Laplacian Eigenmaps (MR-LEM) on a dataset of transformed faces images similar to the 2D shapes dataset used in Higgins et al. (2017). Rather than the combination of 3 shapes used there, we take a single 32x32 pixel grayscale image of a face and apply similarity-preserving transformations - translation, rotation, and scaling - to form a dataset of 64x64 pixel images. We tried both exhaustively enumerating different possible transformations over all positions, angles and scales at some resolution, and also sampling the transformations from a Gaussian distribution, so that the underlying distribution over latents would not have a unique factorization structure. We use the ordinary adjacency matrix with $A_{ij} = 1$ if $i$ is in the 20 nearest neighbors of $j$ or vice versa. We compute the bottom 10 eigenvalues and filter with $\epsilon = 0.5$. In both cases we find MR-LEM can mostly disentangle translation and rotation, though no dimension of the embedding seems to be correlated with scale. Results of MR-LEM applied to both faces dataset are shown in Figs. 3 and 4 in the Appendix, along with comparisons to the Controlled Capacity Increase VAE (Burgess et al., 2017). Interestingly, MR-LEM is able to discover the true circular topology of the rotation factor, while CCI-VAE collapses this factor along a line.

We also evaluated MR-LEM on the same subset of NORB as Hadsell et al. (2006) - a single instance of the "airplane" category across all lighting, azimuth and elevation conditions. Note that this is only 972 data points, a relatively small dataset by current standards. Because different lighting conditions tend to cluster together, we used a kernel that tapers roughly linearly with distance. Say node $i$ is the $n_{i \to j}$th nearest neighbor of node $j$, then let $q_{i \to j} = \frac{1}{n_{i \to j}}$ if $n_{i \to j} \le k$ and 0 otherwise. Then we construct the Gram matrix to be $A_{ij} = q_{i \to j} + q_{j \to i}$. This matrix is sparse, symmetric, weights nearer neighbors more heavily, and is more robust to changes in the relative distance between points in different parts of the latent space than the RBF kernel. In our experiments we used $k = 200$ neighbors, $\epsilon = 0.3$ and computed the 50 bottom eigenvectors before filtering. Results are shown in Fig. 5, including a comparison against baseline Laplacian Eigenmaps, along with a longer discussion in the appendix. Note that the degenerate eigenvectors are filtered out, and azimuth, elevation and lighting seem to either be encoded by different dimensions or hierarchically within the same dimension.

We also investigated using the embedding found by MR-LEM for classification on NORB. We classified an element of the test set as the class of its nearest neighbor of the training set - a naive classification method which achieves an error rate of 18.4% from raw pixels and 16.6% from the 95 first principal components (LeCun et al., 2004). Using Laplacian eigenmaps without filtering out redundant eigenvectors, we achieve an error rate of 15.6% with 95 embedding dimensions, and with $\epsilon = 0.1$, MR-LEM gives a classification rate of 13.8%. Further results with different hyperparameters are given in Fig. 6. While this is still far short of the current state-of-the-art of 1.4% using capsule networks (Hinton et al., 2018), this shows that even a naive classification method can be improved by choosing an embedding carefully.

## 4    DISCUSSION

One downside of the proposed method is the requirement that a large number of eigenvectors be precomputed then filtered. Ideally the predictable eigenvectors could be filtered out during the process of eigenvector computation itself. However in practice, the computation of even approximate nearest neighbors is significantly slower than one iteration of an eigensolver for large datasets, meaning filtering during learning with this approach is probably not practical.

We have shown that only a small modification to a classic algorithm is sufficient to reproduce some of the results from recent deep learning architectures. While VAEs struggle to learn the true topology of the latent space, Laplacian Eigenmaps is capable of recovering this, all without having to learn a generative model. We hope this will inspire researchers to revisit classic algorithms as baselines and motivate closer investigation into the reasons behind common failure modes of these algorithms.

ACKNOWLEDGMENTS

Thanks to Kimberly Stachenfeld, Danilo Rezende, Irina Higgins and Raia Hadsell for helpful discussions.

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

## A    ILLUSTRATION OF EIGENVALUE SCALING ON GRID-STRUCTURED DATA

In Fig. 1 we give an example illustrating the intuitive argument in Section 1.2. We plotted the bottom eigenvectors of the graph Laplacian for an evenly-spaced grid (24 by 25, the slight difference is to break symmetry to avoid degenerate eigenvalues) and an unevenly-spaced grid (11 by 41). To recover an embedding that is aligned with the natural coordinate system for the grid, the bottom two eigenvectors of the evenly spaced grid will work, but not for the unevenly spaced grid: the first 3 Fourier modes in the $x$ direction all have smaller eigenvalue than the lowest mode in the $y$ direction. Instead, the 1st and 4th eigenvectors would make a natural coordinate system for the graph.

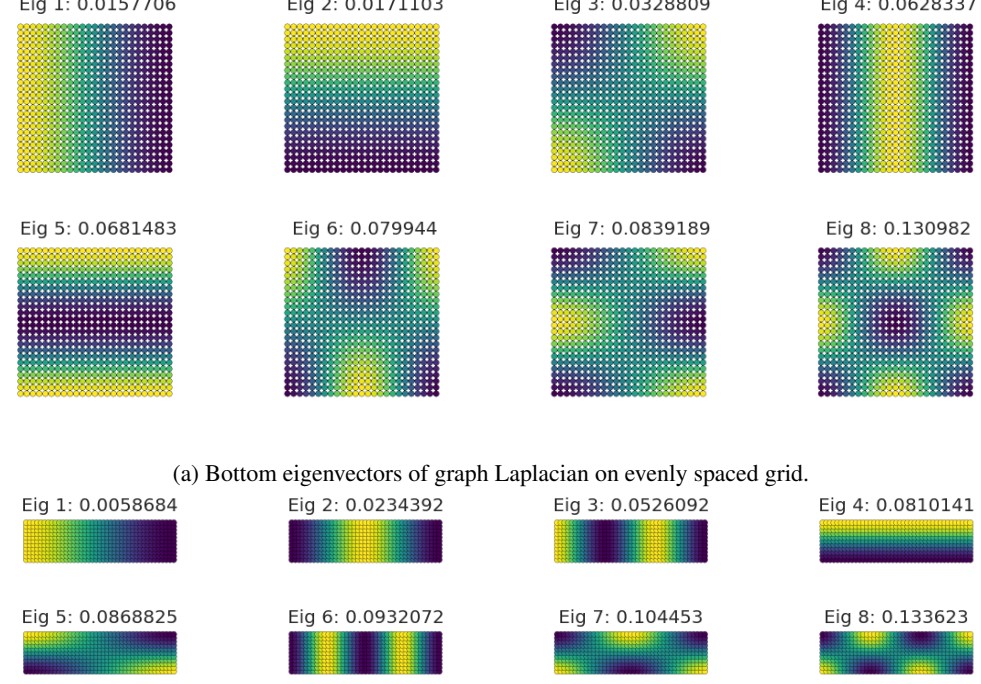

(a) Bottom eigenvectors of graph Laplacian on evenly spaced grid.

(b) Bottom eigenvectors of graph Laplacian on unevenly spaced grid.

Figure 1

## B    DETAILS OF SIMILARITY-TRANSFORMED FACES

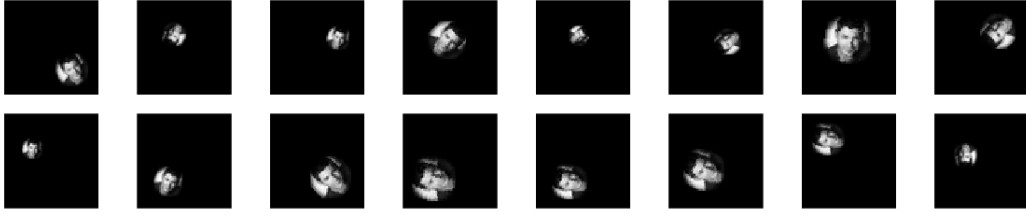

Figure 2: Samples from Similarity-Transformed Faces

The similarity-transformed faces dataset was formed by applying translation, scaling and rotational transforms to a single 32x32 face image on a 64x64 canvas. To avoid loss of information during rotation, the image was circularly cropped. 32 different translations in $x$ and $y$ were applied, along with 40 evenly-spaced rotations and 10 different scales between 16 and 32 pixels in size, for a total dataset of 409600 images. Examples are shown in Fig. 2. For the dataset with Gaussian-distributed

latents, 100000 images were sampled. Positions of the center of the image in $x$ and $y$ were sampled from $\mathcal{N}(32, 3.2)$, rotation in degrees was sampled from $\mathcal{N}(0, 60)$ and scale in pixels was sampled from $\mathcal{N}(24, 2.5)$.

## C    CONTROLLED CAPACITY INCREASE VARIATIONAL AUTOENCODER

The Controlled Capacity Increase-Variational Autoencoder (CCI-VAE) (Burgess et al., 2017) is an extension of variational autoencoders for disentangling latent factors of variation, in the spirit of $\beta$-VAE (Higgins et al., 2017). With the $\beta$-VAE good latent disentangling often comes at the expense of not representing some factors in a dataset, and reduced reconstruction and sample quality. However, CCI-VAE can generally obtain more robust disentangling, with all factors in a dataset encoded (and correspondingly high-quality reconstructions and samples). In CCI-VAE, the coding capacity of the latent bottleneck is gradually increased during training, which encourages the factors of variation in the data to be encoded progressively and into separate latent dimensions (according to their importance – or scale – for reconstruction). To achieve this, CCI-VAE modifies the KL divergence term in the standard VAE training objective:

$$\mathcal{L}(\theta, \phi; \mathbf{x}, \mathbf{z}, C) = E_{q_\phi(\mathbf{z}|\mathbf{x})}[\log p_\theta(\mathbf{x}|\mathbf{z})] - \gamma \left| q_\phi(\mathbf{z}|\mathbf{x}) p(\mathbf{z}) - C \right| \tag{1}$$

Under this objective, the deviation of the KL from a target value $C$ is strongly minimized (with $\gamma$ modulating the strength of this penalty, set to 80 in our experiments). We linearly increased C from 0.5 to 25.0 nats over the course of 450,000 training iterations in our experiments.

The neural network used for the CCI-VAE was as follows. An encoder consisting of 4 convolutional layers, each with 32 channels, 4x4 kernels, and a stride of 2, followed by 2 fully connected layers, each of 128 units. The latent distribution consisted of one fully connected layer of 20 units parameterizing the mean and log standard deviation of 10 Gaussian random variables. The decoder architecture was the transpose of the encoder, but with the output parameterizing Bernoulli distributions over the pixels. ReLU activations were used throughout. The optimiser used was Adam (Kingma & Ba, 2014) with a learning rate of 4e-4.

## D    COMPARISON OF CCI-VAE AND MR-LEM ON
## SIMILARITY-TRANSFORMED FACES

To make sure that the results weren't too strongly influenced by the choice of distribution over latent states, we compared MR-LEM and CCI-VAE on similarity-transformed faces both where latents were enumerated over a grid (Fig 3) and sampled from a Gaussian (Fig 4). In the grid-enumerated case, MR-LEM is clearly able to disentangle translation and rotation (though there is some weak translation dependence in dimensions 3 and 4 and weak rotation dependence in dimensions 1 and 2). Interestingly, the learned embeddings for translation are cleanly arranged into a square, and the learned embedding for rotation form a clean circular shape. Oddly, there is no significant correlation with scale along any dimensions. This may be partly due to the fact that there were significantly fewer scales present in the dataset. However, MR-LEM applied to the "true" graph structure of the latents was able to recover all directions. Thus we suspect that the absence of scale has something to do with the choice of kernel for constructing the adjacency matrix, which we intend to explore further.

The CCI-VAE is able to successfully disentangle the different latent factors, but it is somewhat sloppy in the learned embedding relative to MR-LEM. This is likely due to the Gaussian prior that most VAEs use. Rotation in particular is not mapped to a circle, but is bisected in half and mapped to a single dimension. In some experiments an extraneous rotation-coding dimension appeared, but it still did not find a rotationally symmetric structure.

Results with latents sampled from a Gaussian distribution are given in Fig. 4. Here the data more closely match the prior of the CCI-VAE, and the learned embeddings closely map to a Gaussian distribution. MR-LEM is still mostly able to disentangle translation and rotation, with some hierarchical encoding of $y$ position and rotation, as in the encoding of lighting and elevation in NORB.

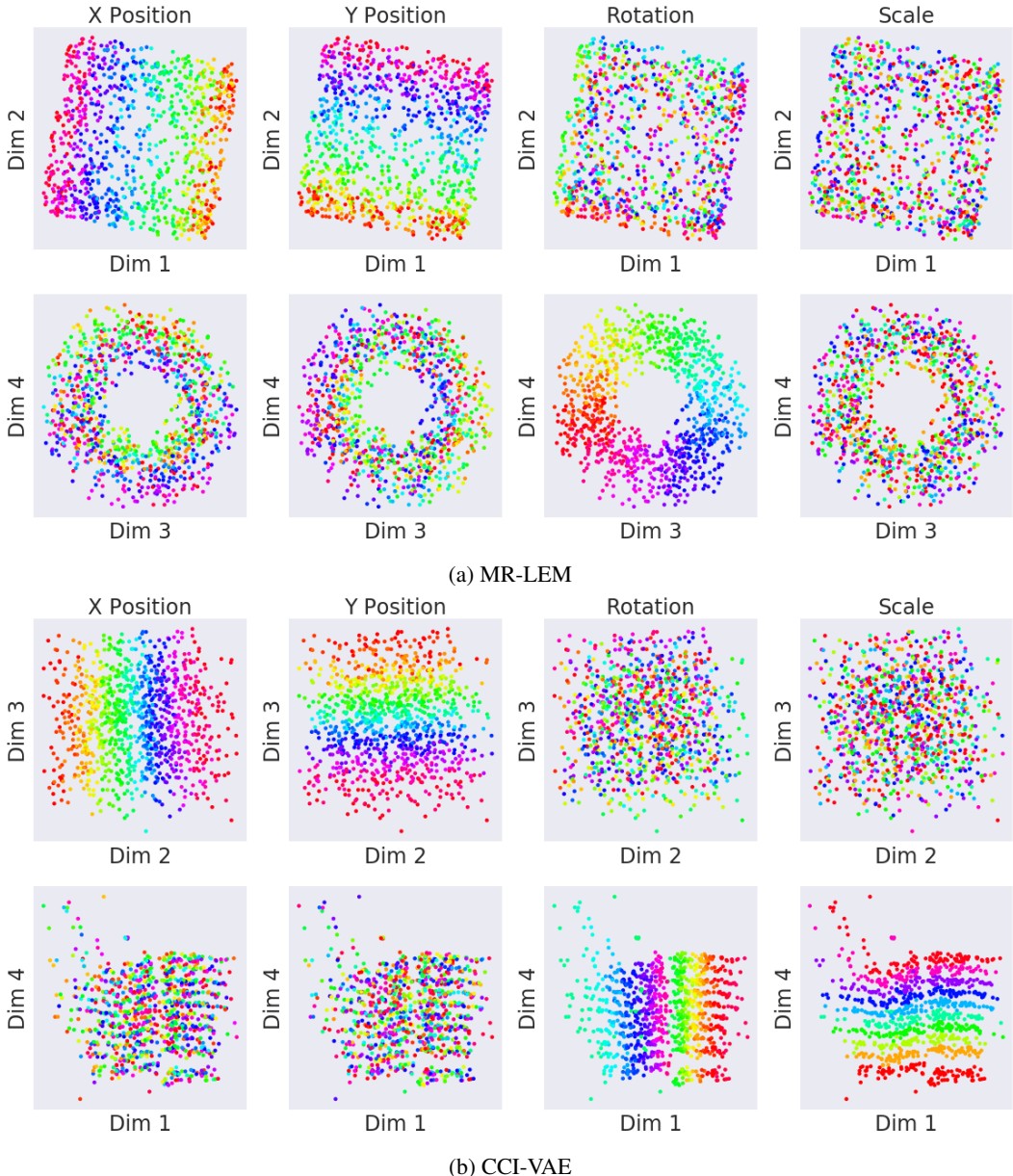

Figure 3: Results of MR-LEM and CCI-VAE applied to similarity-transformed faces, where the latent variables are enumerated across a grid. Points in each column are color-coded according to the true value of the latent variable in the title.

The first dimension primarily correlates with rotation, followed by $y$ position, while the second dimension correlates with $x$ position and the third with $y$ position. The ability to disentangle $x$ and $y$ despite being sampled from an isotropic Gaussian is likely due to anisotropy in the object being translated - a shift in the $x$ direction causes a different change in pixel space than a shift in the $y$ direction. We also suspect the slight entangling of $y$ position between the first two dimensions in CCI-VAE is due to the anisotropy of the face and would be ameliorated in a dataset with a large ensemble of faces or other objects.

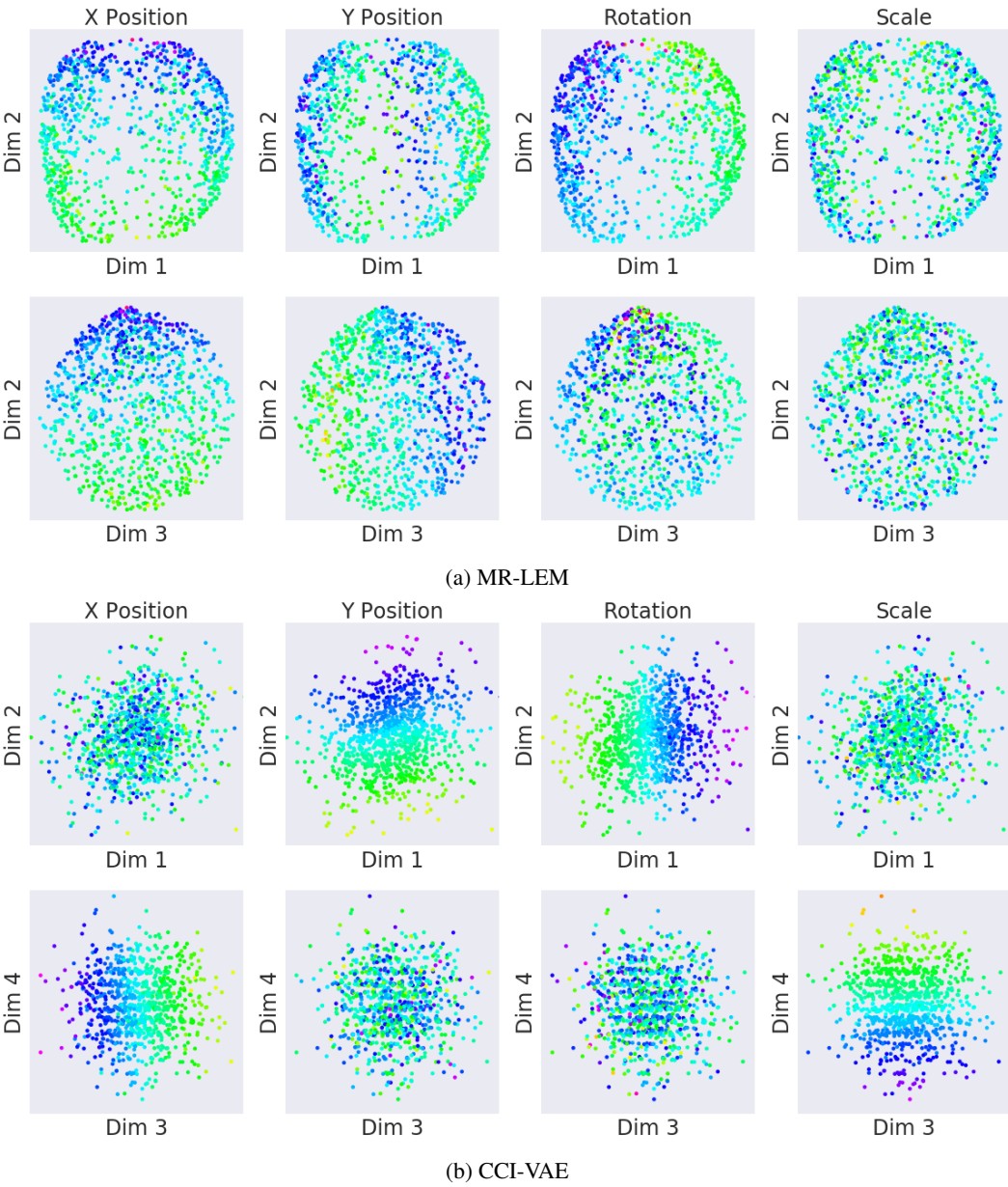

Figure 4: Results of MR-LEM and CCI-VAE applied to similarity-transformed faces, where the latent variables are sampled from a Gaussian distribution.

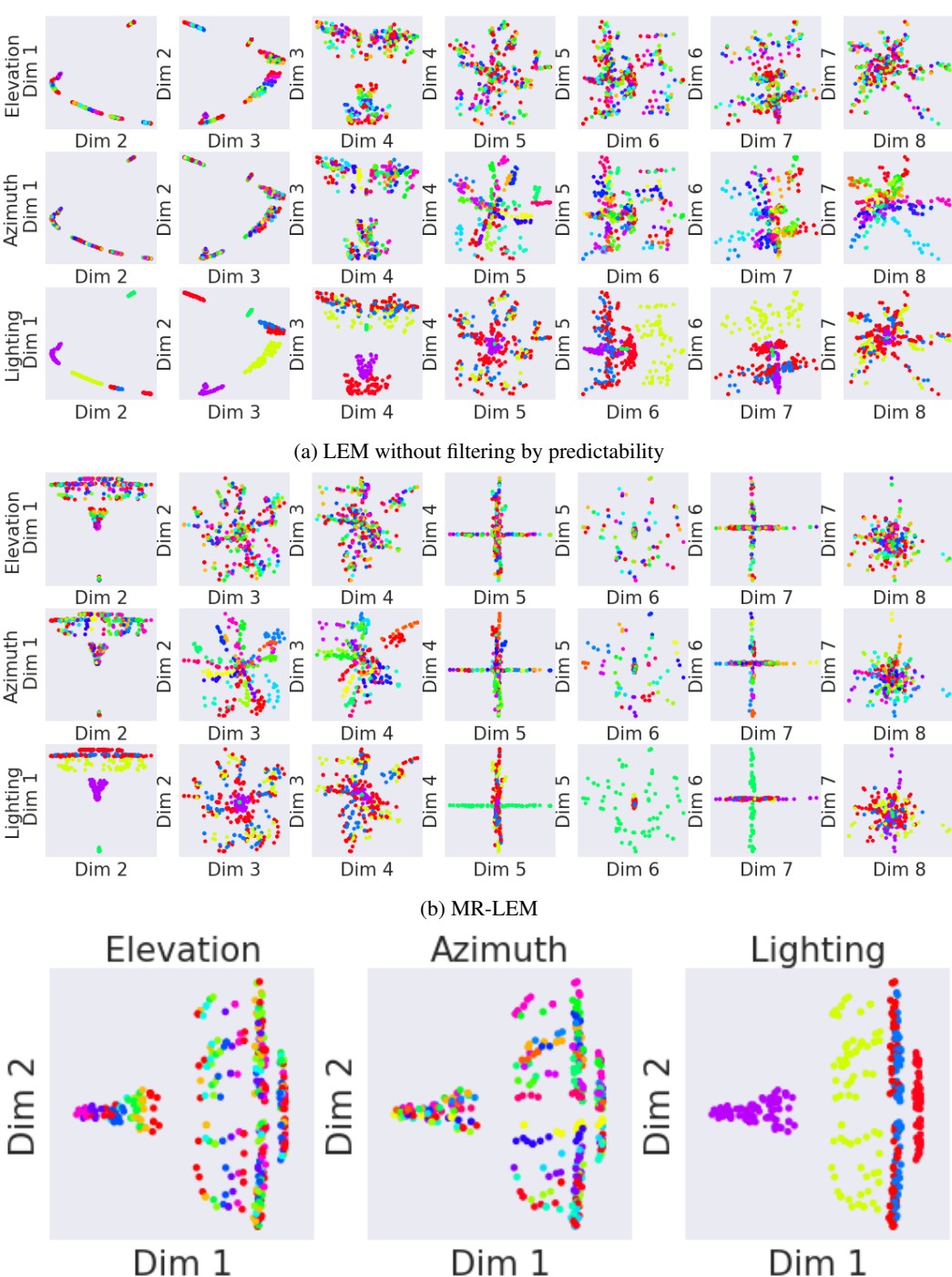

(a) LEM without filtering by predictability

(b) MR-LEM

(c) Detail of first two dimensions on MR-LEM highlighting hierarchical disentangling

Figure 5: Results of LEM and MR-LEM applied to one instance of the airplane category under different lighting and pose conditions. All 2D projections of consecutive dimensions are shown for the first 8 dimensions, with the different latent variables plotted by color in different rows.

# E LAPLACIAN EIGENMAPS ON NORB

Laplacian Eigenmaps without filtering predictable dimensions (Fig. 5a) is consistent with the results of Hadsell et al. (2006) using LLE: the first two embedding dimensions are collapsed along a thin curve. Dimension 6 also re-encodes lighting and azimuth in a slightly different way from dimension 1. This is exactly the outcome that would be expected in a dataset where different factors of variation are scaled differently in observation space. Filtering eigenvectors to minimize predictability (Fig. 5b) significantly reduces the redundancy of the learned embedding.

Note that Laplacian eigenmaps with this choice of kernel hierarchically encodes *both* lighting and elevation in the first dimension - as lighting conditions are quite different, the embeddings naturally cluster together. In Fig. 5c we show a detail of the first two dimensions of the embedding, so that this hierarchical encoding is clearer. One lighting condition in particular (in green) is reduced almost to a single point. Only in dimensions 5 and 6 of the embedding are the different points under this lighting condition mapped out to different points in latent space. These two dimensions correspond to the 23rd and 26th smallest nontrivial eigenvalues of the graph Laplacian, which shows how important it is to filter out predictable dimensions to discover a useful embedding. Also note that elevation is almost completely disentangled from azimuth. The former is encoded along the first dimension and the radial direction of dimensions 2 and 3, while the latter is encoded in the angular direction of dimensions 2 through 4. This demonstrates the importance of choosing minimally predictable eigenvectors as the embedding.

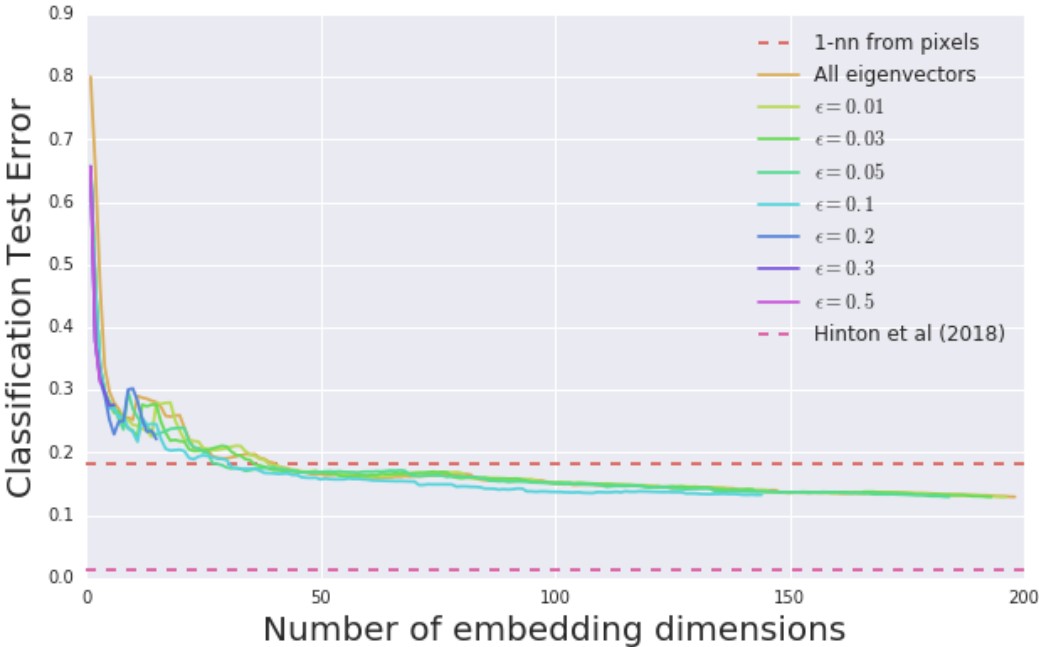

Figure 6: Classification performance of Laplacian eigenmaps and MR-LEM on NORB.

