# OpenReview forum: "Minimally Redundant Laplacian Eigenmaps"
_ICLR.cc/2018/Workshop — Accept_

### Official Review · AnonReviewer2 · 2018-03-04
**Authors of this paper demonstrated that the classic nonparametric manifold learning methods often suffer from the issues of collapsing embeddings, and argued that this collapse is due to the method by which embedding vectors are chosen. To fix this issue, minimally redundant laplacian eigenmaps were proposed.**

**Rating:** 6
**Confidence:** 4

**Review:**

Authors of this paper demonstrated that the classic nonparametric manifold learning methods often suffer from the issues of collapsing embeddings, and argued that this collapse is due to the method by which embedding vectors are chosen. To fix this issue, minimally redundant laplacian eigenmaps were proposed.

Authors stated that, due to the different changes to the latents cause different scales of changes in observation space, the nearest neighbors graph is more elongated along certain dimensions than others.
1. It is not clear why selecting appropriate eigenvectors can overcome this issue. My concern is that the eigenvectors from Laplacian eigenmaps with incorrect nearest neighbor graph might be also incorrect.

2. How about normalizing the scale of the observation data first and then performing Laplacian eigenmaps?

From the description of selecting strategy in Section 2, it is not clear how to select unpredictable vectors. It seems that the method greedily selects one vector from the rest of vectors by setting the second smallest as the initial one. However, experimental results demonstrated certain combination of any two vectors from Laplacian eigenmap. More details are needed for better understanding.

---

### Official Review · AnonReviewer1 · 2018-03-08
**Minimally Redundant Laplacian Eigenmaps**

**Rating:** 7
**Confidence:** 3

**Review:**

- The authors propose a new method called Minimally Redundant Laplacian Eigenmaps which starts from a (standard) spectral clustering with normalized graph Laplacian and then, instead of directly using the eigenvectors, consider a new criterion for selecting few eigenvectors for a low-dimensional embedding. A good motivation (scaling issues) is given for applying this modified procedure. The authors describe the new criterion and show an extensive experimental section with additional material in appendix. Overall, they make a nice and valuable contribution at this point.

- On the other hand, a possible drawback of the proposed method might be that one does not known to which underlying objective the final solution is corresponding, after the two-steps procedure (because implicitly the final solution doesn't correspond to the original objective of step 1).

In order to obtain few components directly from the problem formulation in the first step, one would probably need to modify the objective function (or have a different variational principle) in step 1. The following recent work achieves few components directly from the problem formulation:

Fanuel M., Aspeel A., Delvenne J.C., Suykens J.A.K., Positive semi-definite embedding for dimensionality reduction and out-of-sample extensions, arXiv:1711.07271

- In section 3 the authors choose k=200 and epsilon =0.3. Are the result sensitive with respect to these choices? (ideally it would be good if a model selection procedure could be given).

---

### Decision · Program_Chairs · 2018-03-20
**ICLR 2018 Workshop Acceptance Decision**

**Decision:**

Accept

**Comment:**

Congratulations, your paper was accepted to the ICLR workshop.